# Genome-wide identification and expression analysis of late embryogenesis abundant protein-encoding genes in rye (*Secale cereale* L.)

**Mengyue Ding**[1☯]**, Lijian Wang**[1,2☯]**\*, Weimin Zhan**[1]**, Guanghua Sun**[1]**, Xiaolin Jia**[1]**, Shizhan Chen**[1]**, Wusi Ding**[1]**, Jianping Yang**[1]**\***

**1** College of Agronomy, Henan Agricultural University, Zhengzhou, China, **2** Department of Criminal Science and Technology, Henan Police College, Zhengzhou, China

☯ These authors contributed equally to this work.

\* jpyang@henau.edu.cn (JY); wanglijian2012@163.com (LW)

**Data Availability Statement:** All rye RNA-Seq files are available from the NCBI Sequence Read Archive Database (accession number SRP293835).

## Abstract

Late embryogenesis abundant (LEA) proteins are members of a large and highly diverse family that play critical roles in protecting cells from abiotic stresses and maintaining plant growth and development. However, the identification and biological function of genes of *Secale cereale LEA* (*ScLEA*) have been rarely reported. In this study, we identified 112 *ScLEA* genes, which can be divided into eight groups and are evenly distributed on all rye chromosomes. Structure analysis revealed that members of the same group tend to be highly conserved. We identified 12 pairs of tandem duplication genes and 19 pairs of segmental duplication genes, which may be an expansion way of *LEA* gene family. Expression profiling analysis revealed obvious temporal and spatial specificity of *ScLEA* gene expression, with the highest expression levels observed in grains. According to the qRT-PCR analysis, selected *ScLEA* genes were regulated by various abiotic stresses, especially PEG treatment, decreased temperature, and blue light. Taken together, our results provide a reference for further functional analysis and potential utilization of the *ScLEA* genes in improving stress tolerance of crops.

## Introduction

Plants are affected by a variety of biotic and abiotic stresses during their lifetime due to their inability to escape harmful environmental conditions. Late embryogenesis abundant (LEA) proteins are highly hydrophilic proteins with antioxidant and ion binding properties that function in membrane and protein stabilization, osmotic regulation, and as a protective buffer against dehydration [1]. Thus, they play important roles in protecting cells from abiotic stress, and in normal growth and development. LEA proteins were first isolated from cotyledons of cotton (*Gossypium hirsutum* L.) at late embryonic development stages [2], and in recent decades, have also been found in species from algae to angiosperms, such as cyanobacteria [3], *Arabidopsis thaliana* [4], rice (*Oryza sativa* L.) [5], wheat (*Triticum aestivum* L.) [6], maize (*Zea mays* L.) [7], and in prokaryotes and invertebrates, such as rotifers [8,9]. In plants, *LEA*

**Funding:** This study was funded by the National Natural Science Foundation of China to JP (General Program, 31871709).

**Competing interests:** The authors have declared that no competing interests exist.

genes are highly expressed at the late stage of seed maturation [2] and LEA proteins accumulate in roots, stems, and other organs throughout the plant growth duration [10].

In higher plants, based on the similarity of amino acid sequences and differences in conserved domains, the LEA family is generally divided into eight groups: LEA1, LEA2, LEA3, LEA4, LEA5, LEA6, Dehydrin, and seed maturation protein (SMP) [11]. In recent studies abscisic acid (ABA) stress ripening (ASR) proteins were classified as LEA7 group proteins [12,13], extending the LEA family to nine groups. The division of LEA family members varies among species [14]. Most *LEA* genes encode small proteins with molecular weights ranging from 10 to 30 kDa, and they are predominantly composed of a repeating arrangement of hydrophilic amino acids that form a highly hydrophilic structure [15]. Most LEA proteins have low hydrophobicity and a large net charge, which allow them to be "completely or partially disordered" [16,17], these proteins may act as a novel form of molecular chaperone, or 'molecular shield', to help prevent the formation of damaging protein aggregates during water stress [18]. Each group of LEA proteins has its own unique conserved motif [19]. For example, LEA1 proteins contain a 20-amino-acid motif (GGETRKEQLGEEGYREMGRK) [20]. Dehydrin proteins contain a motif called the K-segment (EKKGIMDKIKEKLPG) [21,22]. LEA4 proteins have a conserved domain of 11-amino-acid sequences (TAQAAKEKAGE) [21]. These conserved sequences (motifs) have been preserved through long-term evolution, and play an important role in plant responses to environmental stress.

Previous studies have shown that LEA proteins are ubiquitous and localized in the cytoplasm, nucleus, chloroplast, mitochondria, and endoplasmic reticulum [23]. Thus, not all LEA proteins are localized in the same part of the cell, and that their particular functions depend on their intra-cellular locations. The expression of LEA proteins is often induced by abiotic stresses such as drought, heat, cold, and exogenous hormone at different development stages and tissues of plants [1,4]. Moreover, overexpression of the *LEA* genes can improve stress tolerance of transgenic plants. For example, overexpression of the barley (*Hordeum vulgare* L.) *HVA1* gene promoted drought and salt stress tolerance in rice [24] and wheat [25]. Overexpression of the wheat *DHN-5* enhanced osmotic stress tolerance in *Arabidopsis* [26], and activated pathogenesis-related protein expression [27]. The above studies demonstrate that the *LEA* genes could potentially be used to improve the abiotic stress tolerance of crops.

Rye (*Secale cereale* L.) is a diploid Triticeae species closely related to wheat and barley, and an important crop for feed and food [28]. Rye has excellent disease and stress resistance, and it has been used as a source to improve wheat resistance to pathogens in the past few decades [29]. Rye-wheat chromosome translocation has been widely used in wheat breeding around the world, and translocation lines have notably increased yield [30,31]. Studies on LEA proteins in wheat and barley have been reported [6,25]. However, the distribution and function of LEA proteins in rye have been rarely reported [32]. In this study, we identified 112 *ScLEA* genes based on the rye genome sequence. The phylogenetic analysis, structural characterizations, evolutionary relationships, and expression profiles of the *ScLEA* genes were systematically analyzed. These results will contribute to our understanding of the LEA family in rye and to the utilization of these *LEA* genes in homologous or heterologous systems.

## Materials and methods

### Identification of *LEA* genes in the rye genome

The nucleotide and protein sequences, as well as the gene annotation files, were downloaded from the rye genome database (http://pgsb.helmholtz-muenchen.de/plant/rye/index.jsp). The HMM profiles (LEA1: PF03760, LEA2: PF03168, LEA3: PF03242, LEA4: PF02987, LEA5: PF00477, LEA6: PF10714, Dehydrin: PF00257, SMP: PF04927) were downloaded from the

Pfam database (http://pfam.sanger.ac.uk/). Then, HMMER 3.0 was used to search the encoded protein sequences with the default parameters and a filter threshold of 0.01. In addition, the obtained amino acid sequences were used as queries in searches against both the CDD database (https://www.ncbi.nlm.nih.gov/cdd/), and SMART database (http://smart.embl-heidelberg.de/), and the repeated or non-LEA domain sequences were eliminated manually. Finally, the identified rye LEA proteins were named according to the group and gene ID order. The physical and chemical properties of ScLEA proteins were analyzed using the Prot-Param online tool (https://web.expasy.org/protparam/).

## Phylogenetic analysis of LEA proteins in rye

To investigate the evolutionary relationships of LEA proteins in rye, multiple alignments of the full-length protein sequences were performed using MAFFT [33] (http://mafft.cbrc.jp/alignment/software/) with the default parameters. A phylogenetic tree was constructed based on these alignments using the Neighbor-Joining (NJ) method with a bootstrap test of 1000 replicates for assessing internal clade reliability using MEGA X software [34]. The Evolview website (https://www.evolgenius.info/evolview/#login) was used to modify the phylogenetic tree.

## Analysis of gene structures and conserved domains of *LEA* genes in rye

The exon-intron structures of the *ScLEA* family genes were determined by aligning the coding sequences with the corresponding genomic sequences, and visualized using the online software GSDS (http://gsds.gao-lab.org/index.php). Conserved domains of the ScLEA proteins were predicted using MEME (http://meme-suite.org/index.html).

## Distribution of *LEA* genes on rye chromosomes

*ScLEA* genes were mapped on rye chromosomes according to the positional information from the rye genome annotation database, and the chromosome physical location map was displayed using Mapchart [35] software. According to Holub [36], chromosomal region of 200 kb containing two or more genes are defined as a tandem duplicated event. The Multiple Collinearity Scan toolkit (MCScanX) [37] with default parameters was used to analyze gene duplication events.

## Expression profile analysis of rye *LEA* genes

The rye RNA-Seq datasets used for generating gene expression levels were downloaded from the NCBI Sequence Read Archive Database (accession number SRP293835) (https://www.ncbi.nlm.nih.gov/sra/?term=SRP293835). The transcripts of the datasets were aligned, followed by merging and removal of redundant sequences using Hisat (http://ccb.jhu.edu/software/hisat/index.shtml) (version 2.0.4) and Stringtie (http://ccb.jhu.edu/software/stringtie/) (version 1.2.3). The software featureCounts [38] was used to calculate the number of reads and normalized gene expression levels. Expression heatmaps were generated using TBtools software [39].

## Plant materials and stress treatments

For abiotic stress treatments, rye seedlings were grown in hydroponics with a temperature (18˚C) for 10 days, and then treated with 20% (w/v) PEG6000, 200 mM NaCl, 100 mM mannitol, 100 μM ABA, or temperatures of 0˚C or 4˚C. Leaves were collected after 0 h, 3 h, 6 h, 9 h, 12 h, and 24 h. Leaf samples were treated with sterile water and similarly collected as a control. For different light treatments, seedlings were grown in the dark for seven days and subsequently transferred to far-red (FR) light (5 μmol·m$^{-2}$·s$^{-1}$), red (R) light (17.56 μmol·m$^{-2}$·s$^{-1}$),

blue (B) light (13 µmol·m$^{-2}$·s$^{-1}$), or white (W) light (85 µmol·m$^{-2}$·s$^{-1}$) for 4 h. For each sample, three leaves treated in parallel represented three biological replicates. All treated tissue samples were stored at -80˚C for subsequent analysis.

### RNA extraction and gene expression analysis

Total RNA was extracted from the leaves of rye using Eastep® Super RNA Kit (Promega). The GoScript™ Reverse Transcriptase Kit (Promega) was used to reverse- transcribe RNA into cDNA. Quantitative real-time PCR (qRT-PCR) was carried out in the Roche Lightcyler® 480 instrument using SYBR Green (TaKaRa). The *ScActin* gene was used as an internal control. The thermal cycler program used was 95˚C for 10 min, and followed by 45 cycles of 95˚C for 10 s, 60˚C for 10 s, and 72˚C for 20 s. Each reaction was performed in three technical replicates, and the data from qRT-PCR were calculated using the 2$^{-\Delta\Delta Ct}$ method [40]. The significant differences between data were calculated using Student's *t*-test or two-way analysis of variance (ANOVA), and indicated with asterisks (*$P$ <0.05, ** $P$<0.01, *** $P$<0.001). Sequences of the primers used in this study are shown in S1 Table.

### *Cis*-acting element analysis

The Plant CARE website (http://bioinformatics.psb.ugent.be/webtools/plantcare/html/) was used to predict *cis*-acting elements in the regions 1000 bp upstream of the gene start codon and a diagram was visualized using TBtools software [39].

## Results

### Genome-wide identification and phylogenetic analysis of *ScLEA* genes

Based on genome-wide blast searches, 112 *ScLEA* genes were identified from the rye genome database. The identity of the sequences was verified by checking for the presence of conserved domains using the CDD and SMART tools. These 112 genes were divided into eight groups (LEA1–LEA6, Dehydrin, and SMP) based on the phylogenetic analysis (Fig 1). The largest group was LEA2, which contained 56 members, while the smallest groups, LEA4 and LEA6, had only two members each. Groups Dehydrin, LEA1, LEA5, SMP, and LEA3 contained 18, 14, 8, 7, and 5 members, respectively.

The physical and chemical parameters of the 112 ScLEA proteins were calculated using the ProtParam online tool (S2 Table). The smallest ScLEA protein had 78 amino acid residues with a molecular weight of 8.24 kDa (ScLEA5-8) and the largest had 965 amino acid residues with a molecular weight of 99.64 kDa (ScDehydrin-12). The isoelectric point (pI) values ranged from 4.22 (ScSMP-6) to 11.07 (ScLEA2-17), with an average of 8.37. Grand average of hydropathicity (GRAVY) index analysis indicated that most of the ScLEA proteins were hydrophilic (S2 Table). However, the ScLEA2 group was atypical with 67.8% of proteins were predicted as hydrophobic. The aliphatic index reflects the thermal stability of the protein.

Phylogenetic analysis revealed two major clades of the ScLEA family (Fig 1). The LEA2 group proteins were clustered in one clade, while the proteins in the other seven groups formed another clade. According to the evolutionary relationships, one clade including the Dehydrin group (18 numbers) and LEA5 group proteins (8) was closely related to the LEA4 (2), LEA1 (14), and SMP (7) groups, but distant from the LEA6 (2) and LEA3 (5) groups.

### Structural characterization of *ScLEA* genes

The structure of genes is important for determining their expression and function. Structural analysis revealed that the *ScLEA* genes contain few introns; 42% of *ScLEA* genes have one

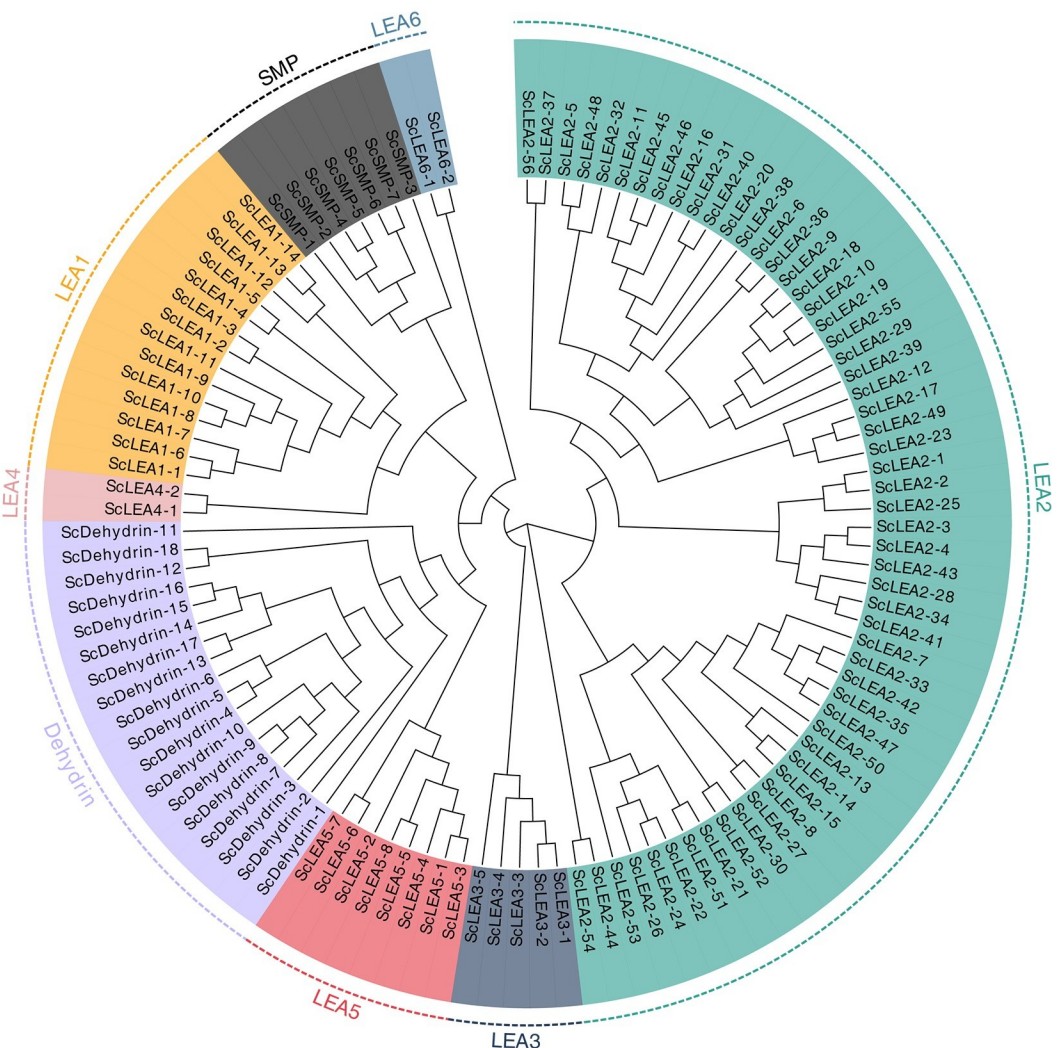

**Fig 1. Phylogenetic analysis of ScLEA proteins.** Multiple sequence alignment of LEA proteins was performed using MAFFT. The phylogenetic tree was constructed using MEGA X software.

intron, 5.4% have two introns, 0.8% have three introns, and 51.8% possess no introns (Fig 2a). There were four intron-free genes and fourteen single-intron genes in the ScDehydrin group. In the ScLEA1 group, seven genes contained no intron, six genes contained one intron, and only one gene contained two introns (*ScLEA1-7*). In the ScLEA2 group, there were 41 intron-free genes, nine single-intron genes, five double-intron genes, and a triple-intron gene (*ScLEA2-37*). There were four free-intron genes and one single-intron gene in the ScLEA3 group. All genes in the ScLEA4, ScLEA5, and ScSMP groups had one intron, and all genes in the ScLEA6 group had no introns. *ScLEA* genes in the same group had similar numbers of exons and introns, indicating that members of the same family tend to be highly conserved.

To elucidate the similarity and diversity of protein domains, each group was separately submitted to MEME for structure analysis. The results showed that each group contained one or more conserved LEA domains (Fig 2b). Moreover, members of the same group were more similar to each other than to members of other groups, indicating that each group of ScLEA proteins may have specific functions.

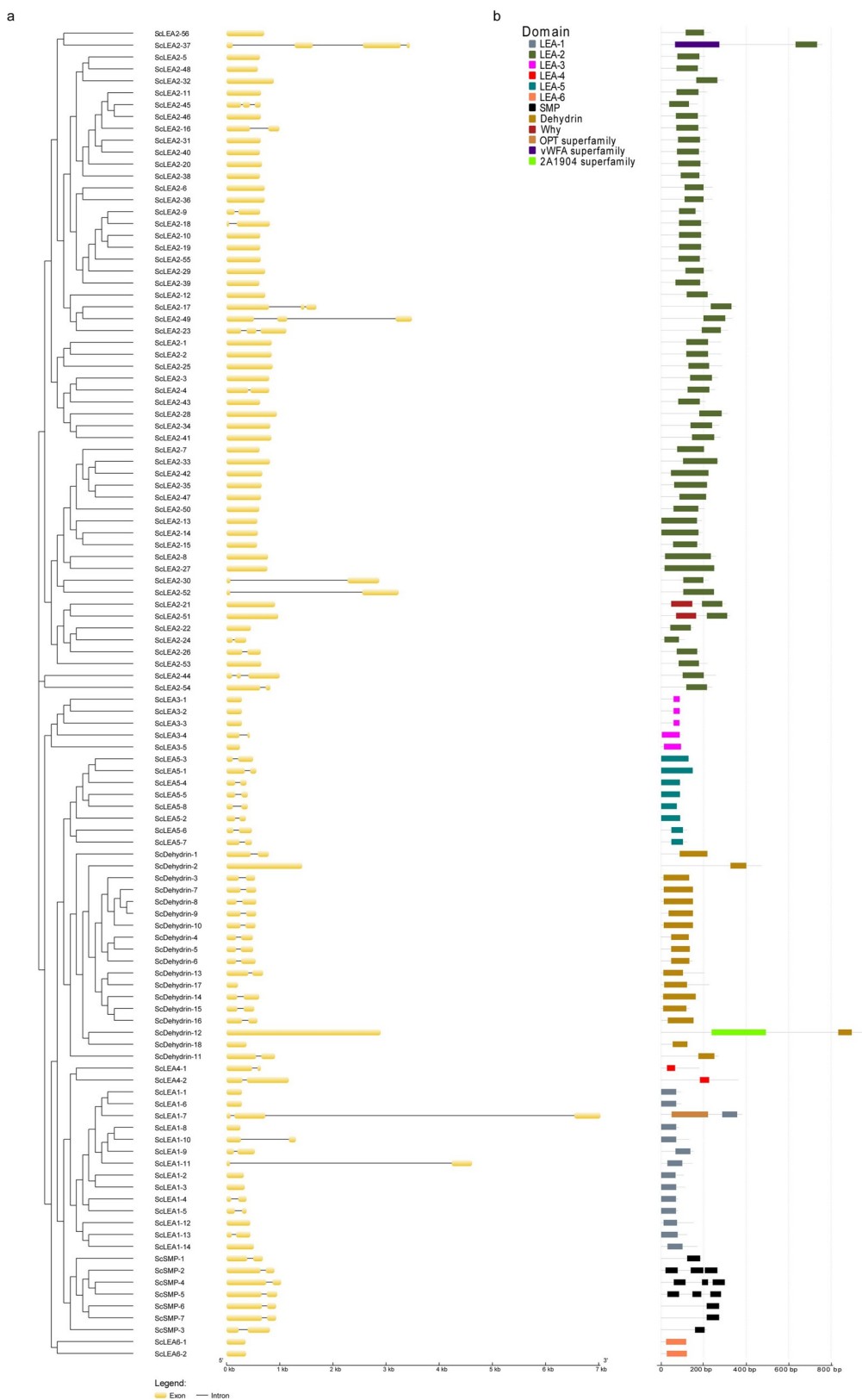

**Fig 2. Schematic representation of exon-intron structures of *ScLEA* genes and conserved domains of ScLEA proteins.** (**a**) The exon-intron structures were examined using the GSDS online tool. The yellow boxes and black lines

represent exons and introns, respectively. **(b)** The ScLEA protein domains were predicted using the MEME online tool.

## Chromosomal locations and gene duplication of *ScLEA* genes

The distribution of *ScLEA* genes was determined by mapping their positions on the rye chromosomes. The 112 *ScLEA* genes were distributed on all seven chromosomes, and their distribution was nearly even among different chromosomes (Fig 3). Chromosome 5 had the most *ScLEA* genes (17.8%, 20 genes), and chromosome 7 had the fewest (12.5%, 14 genes). Chromosomes 1, 2, 3 and 6 each contained 15 genes. Moreover, genes of the largest group, for example, ScLEA2 group, are distributed across all chromosomes to ensure maximum functionalization. However, other LEA groups have a limited distribution and are mainly found on specific chromosomes.

High-density *ScLEA* gene clusters were identified in certain chromosomal regions. Twenty-eight *ScLEA* genes (*ScLEA5-1/2*, *ScLEA5-4/5*, *ScLEA2-9/10*, *ScLEA2-13/14/15*, *ScLEA2-18/19*, *ScLEA1-2/3*, *ScLEA3-2/3*, *ScLEA2-31/32*, *ScSMP-1/2*, *ScDehydrin-7/8/9/10*, *ScDehydrin-12/13*, and *ScDehydrin-14/15/16*) were clustered into 12 tandem duplicated regions on chromosomes 1, 2, 3, 4, 5, and 6 (S3 Table). Chromosome 2 had four clusters, indicating that it is a hot spot for *ScLEA* genes. In addition to the tandem duplication events, 19 segmental duplication events with 27 *ScLEA* genes were also identified using BLASTP and MCScanX (Fig 4 and S4 Table). These results indicate that some *ScLEA* genes are possibly generated by tandem duplication and segmental duplication events.

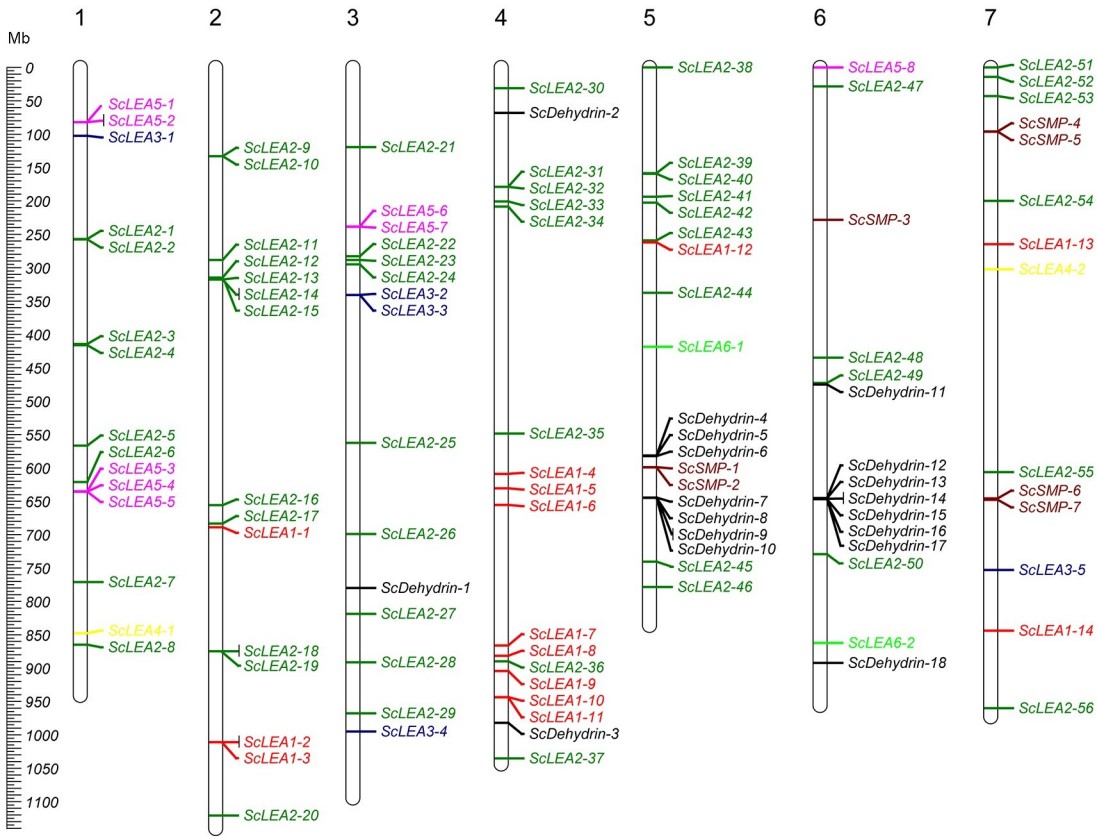

**Fig 3. Distribution of *ScLEA* genes on rye chromosomes visualized using Mapchart software.**

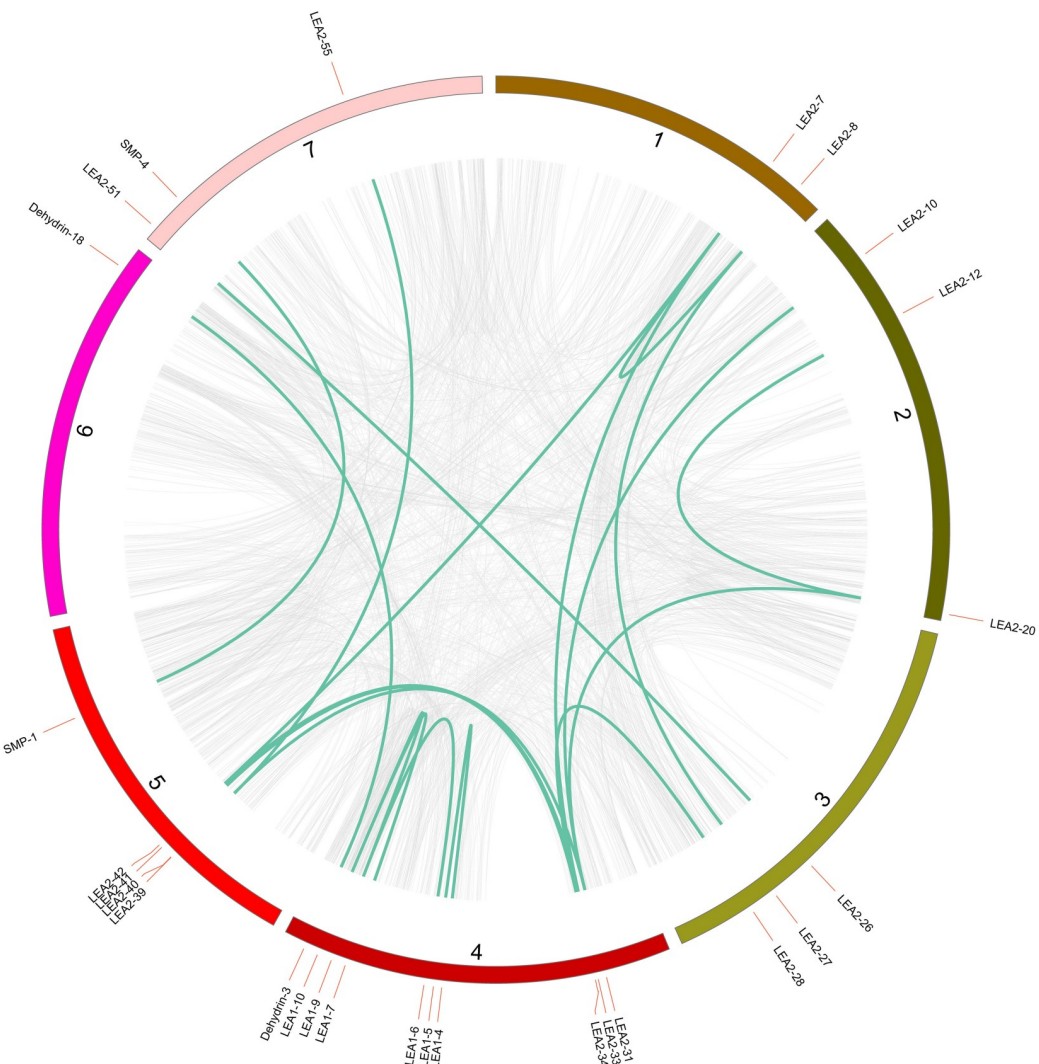

**Fig 4. Schematic representation of the chromosomal distribution and interchromosomal relationships of *ScLEA* genes derived from segmental duplication.** The gray lines indicate all syntenic gene pairs in the rye genome, and the dark turquoise lines indicate segmental duplicated *ScLEA* gene pairs. The number of each chromosome is indicated.

## The expression profiles of *ScLEA* genes

RNA-seq data for different tissues of rye (Weining) (root, stem, leaf, spikelet, and grain) were used to explore the potential biological functions of *ScLEA* genes in growth and development. Expression profiles showed that most *ScLEA* genes were expressed with obvious spatial specificity (Fig 5). *ScLEA2-26* and *ScLEA2-43* were only expressed in spikelets, while *ScLEA2-3*, *ScLEA2-4*, *ScLEA2-27*, *ScLEA2-28*, and *ScLEA2-29* were only expressed in roots. The expression levels of *ScLEA1-3*, *ScLEA1-11*, *ScLEA2-22*, *ScDehydrin-5*, and *ScDehydrin-13* increased continuously during grain development, and reached a maximum at 40 DAA, indicating that these genes may be involved in grain grouting. *ScDehydrin-11*, *ScLEA2-20*, *ScLEA2-31*, and *ScLEA2-51* were highly expressed in different tissues, indicating that they are important at all stages of rye growth and development.

By analyzing the expression profiles of 112 *ScLEA* genes after drought treatment (S1 Fig), we found that the expression levels of *ScDehydrin-10*, *ScDehydrin-11*, *ScDehydrin-13*, and

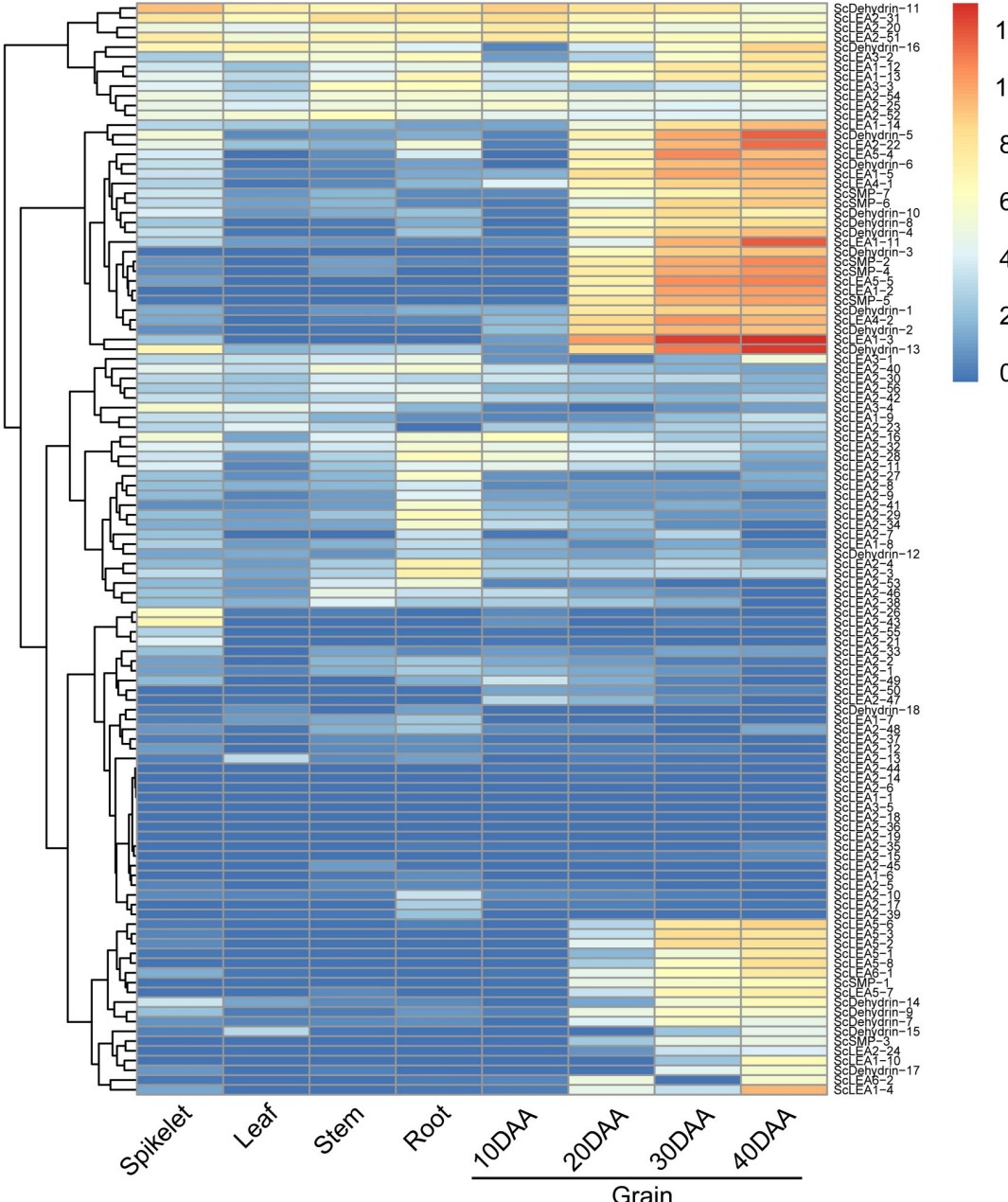

**Fig 5. RNA-seq expression profiles of 112 *ScLEA* genes in different tissues of rye (Weining).** The heatmap was constructed using TBtools. The color scale on the right represents relative expression levels: Red represents high level and blue represents low level.

*ScDehydrin-16* increased the most, followed by *ScLEA1-11*, *ScLEA1-13*, and *ScLEA1-14*, thus, these genes may play major roles in drought stress response. The responses of ScLEA2 group genes were diverse due to its large number of members. For example, *ScLEA2-22* and *ScLEA2-31* respectively showed a significant upregulation and downregulation to drought stress. The responses of the ScLEA5 and ScLEA6 group genes were not obvious, with only the induction of *ScLEA5-5* being prominent.

## The responses of *ScLEA* genes to abiotic stress

To verify the potential roles of *ScLEA* genes in abiotic stress, we randomly selected 12 *ScLEA* genes from eight groups, and their expression in rye leaves after treatment with PEG, ABA, NaCl, or mannitol after 0 h, 3 h, 6 h, 9 h, 12 h, and 24 h were analyzed by qRT-PCR. Similar to the results of RNA-seq under drought treatment (S1 Fig), the results of qRT-PCR showed that the expression of these *ScLEA* genes was induced by different stress treatments (Fig 6 and S2 Fig). Among all treatments, *ScLEA* genes showed the most intense response to PEG, and

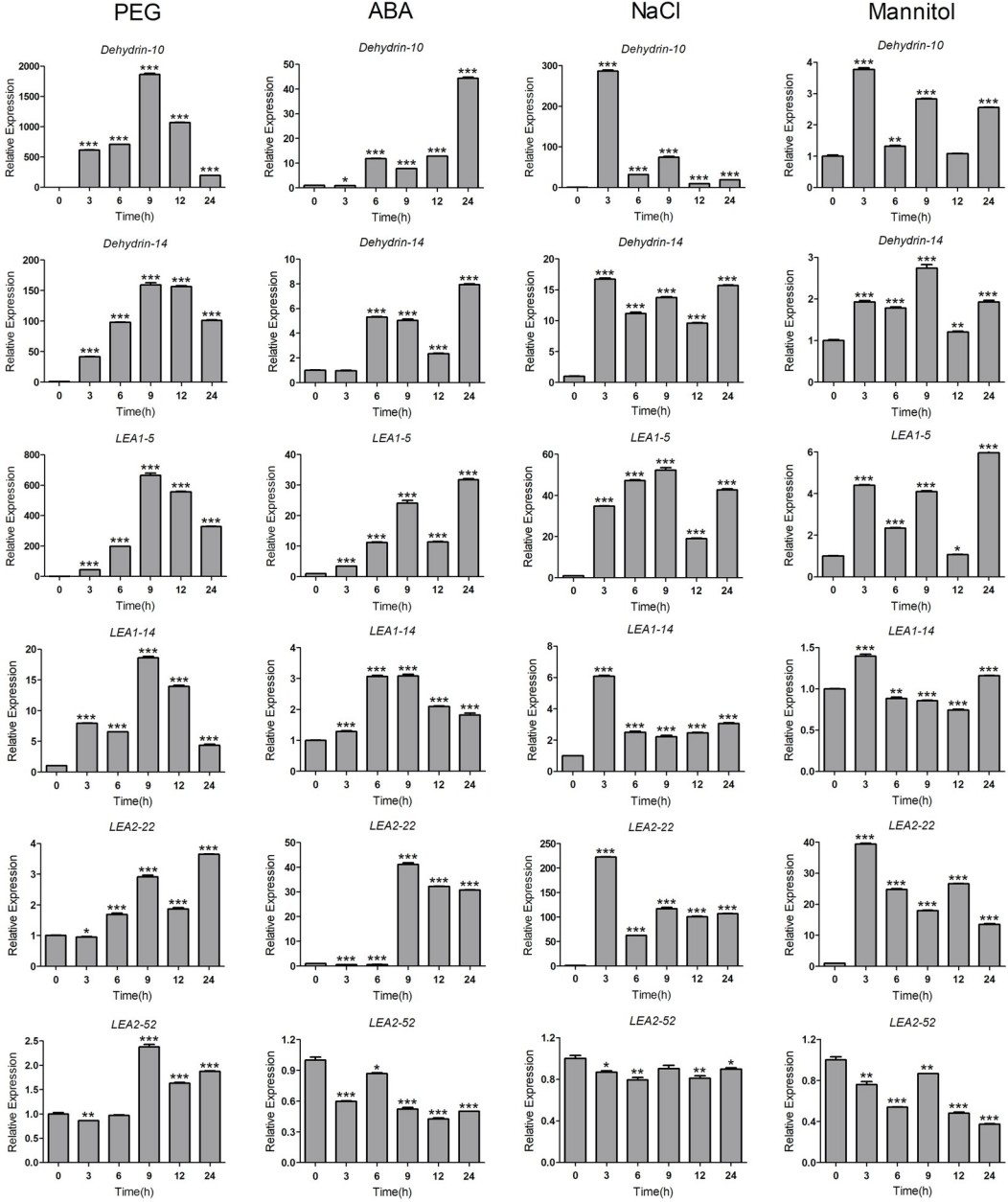

**Fig 6. The relative expression levels of six *ScLEA* genes under different abiotic stresses were analyzed by qRT-PCR.** Ten-day-old seedling leaves were sampled after 0 h, 3 h, 6 h, 9 h, 12 h, and 24 h under 20% PEG6000, 100 μM ABA, 200 mM NaCl, or 100 mM mannitol. The values represent mean ± SEM of three replicates. The significant differences between data were calculated using Student's *t*-test, and indicated with asterisks (*$P < 0.05$, ** $P < 0.01$, *** $P < 0.001$).

compared with the control group, the level of *ScLEA* gene upregulation hundreds of times higher (except for *ScLEA2-22* and *ScLEA2-52*) and lasted for a long time (12–24 h). The 12 *ScLEA* genes did not show the same response to ABA treatment, and some genes were even inhibited (*ScLAE2-22* and *ScLEA3-4*). The expression levels of most genes reached a maximum within 3 h after NaCl treatment, and the subsequent changes were quite different between genes. For some genes, expression was maintained at a continuously high level (*ScDehydrin-14*, *ScLEA1-5*, *ScLEA4-2*, and *ScSMP-4*), while the expression levels of others decreased rapidly (*ScDehydrin-10* and *ScSMP-6*). Under mannitol treatment, only the expression of *ScLEA*2-22 was significantly upregulated and the expression of some genes was inhibited (*ScLEA2-52* and *ScLEA3-4*).

The results of qRT-PCR analysis showed that genes of the ScLEA1, ScLEA2, ScDehydrin, and ScSMP groups were highly sensitive to the four treatments, with significant upregulation of gene expression. Interestingly, the expression of genes in the ScLEA3 group decreased under three treatments (ABA, NaCl, and mannitol), indicating that these treatments repressed gene expression. The above results showed that the gene expression patterns were different under the four treatments, and that the expression levels of genes in each group were also significantly different under the same treatment. The expression of these genes validates the RNA-seq results and also reflects the diversity of *ScLEA* gene functions in response to abiotic stress.

## The response of *ScLEA* genes to cold stress

RNA-seq data analysis showed that *ScLEA* genes were responsive to different durations of low temperature stress treatment (S3 Fig). We analyzed the expression patterns of the same 12 genes under cold stress. *ScLEA* genes expression in rye leaves incubated at 4°C and 0°C for 0 h, 3 h, 6 h, 9 h, 12 h, and 24 h were analyzed by qRT-PCR. The qRT-PCR results showed that most of genes were sensitive to cold stress and showed different degrees of response (Fig 7). The results showed that the expression of *ScLEA* genes increased with increasing duration of cold treatment, and expression was significantly higher than that of the control by 12 h. The responses of *ScLEA* genes to 0°C treatment were significantly stronger than those to 4°C treatment at almost all treatment time points. Among the 12 genes, *ScDehydrin-10*, *ScLEA2-22*, and *ScSMP-6* were strongly responsive to low temperature, and their expression levels were upregulated to hundreds of times compared with the control (0 h). Similar to PEG, ABA, NaCl, and mannitol treatment, the gene expression of *ScLEA2-52* and *ScLEA3-4* to low temperature were inhibited at some time points, suggesting that these genes may be involved in other stress responses, such as biotic stress, or plant growth and development processes. In conclusion, the response of *ScLEA* genes to cold stress reflects the tolerance of rye to low temperatures.

## The response of *ScLEA* genes to photon switching

To elucidate the possible regulatory mechanisms of *ScLEA* genes involved in environmental stress, we analyzed the stress-responsive elements in the promoter regions of 12 *ScLEA* genes (S4 Fig). The ABRE, which is involved in ABA responsiveness, the LTRE, which responds to low temperature, the MBS (MYB binding site) element, which is involved in drought response, the G-box, Box 4 and GT1-motif, which participate in light responsiveness. The ABRE and G-box elements were found in most of the 12 genes, and auxin, salicylic acid, and jasmonic acid responsive elements were also identified.

The responses of 12 *ScLEA* genes to PEG, ABA, NaCl, mannitol, and low temperature were confirmed. The role of *LEA* genes in photon switching response has rarely been reported. To verify whether the *ScLEA* genes respond to photon switching, the expression of *ScLEA* genes

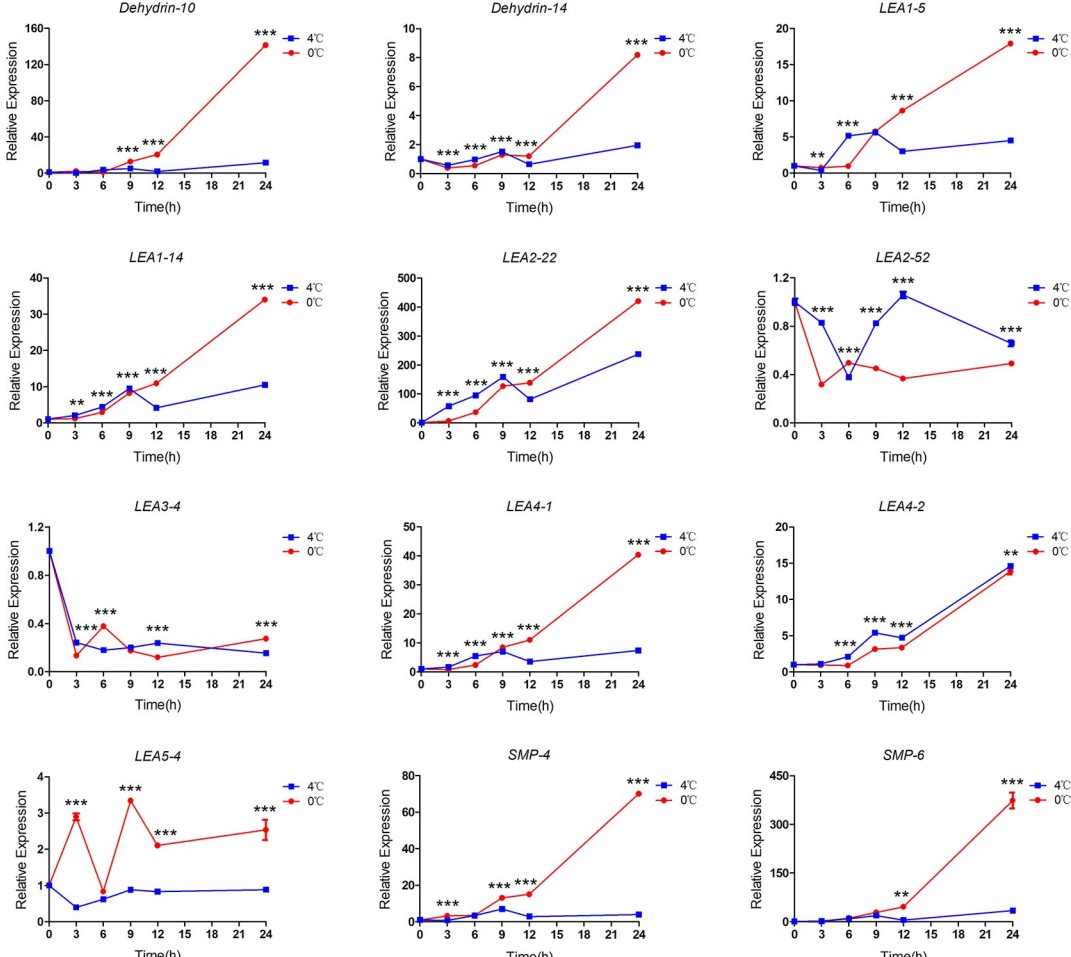

**Fig 7. The relative expression levels of 12 *ScLEA* genes under cold stress were analyzed by qRT-PCR.** Ten-day-old seedling leaves were sampled after 0 h, 3 h, 6 h, 9 h, 12 h, and 24 h at 0˚C or 4˚C. The values represent mean ± SEM of three replicates. The significant differences between data were calculated using two-way ANOVA, and indicated with asterisks (** $P<0.01$, *** $P<0.001$).

in rye leaves treated with FR, R, B and W light for 4 h were analyzed by qRT-PCR (Fig 8). Most of these *ScLEA* genes (*ScLEA1-5*, *ScLEA1-14*, *ScLEA2-22*, *ScLEA2-52*, *ScLEA4-1*, *ScLEA4-2*, *ScDehydrin-10*, and *ScDehydrin-14*) showed obvious responses to blue light. Compared with W light, *ScSMP-4*, *ScLEA4-1*, and *ScLEA4-2* showed a stronger response to R light. The results showed that *ScLEA* genes respond to different types of light, and that the degree of response to the four types of light varied greatly. Thus, plants may express various genes including *LEA* in response to stress.

## Discussion

### Identification and analysis of rye *LEA* genes

*LEA* genes have been reported in many species; however, the genomic identification and annotation of *LEA* genes has not been reported in rye. In this study, 112 *ScLEA* genes were identified, and were divided into eight groups. LEA2, the largest group in the rye family, accounts for about 50% of the total number of family members. The LEA2 group of wheat is also the largest, accounting for 159 of all 281 *LEA* genes [6]. However, the largest group in rice [5] and

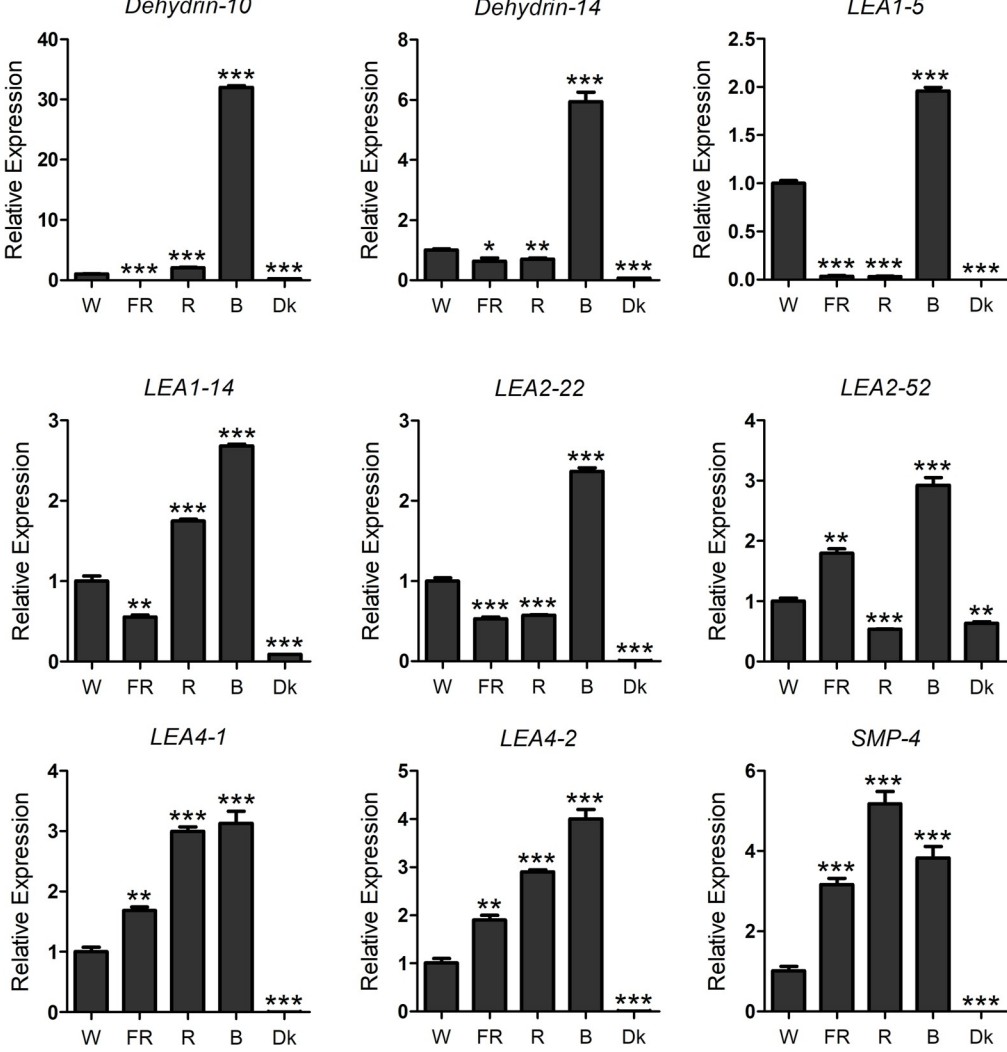

**Fig 8. The relative expression levels of nine *ScLEA* genes under different light conditions were analyzed by qRT-PCR.** Seedlings were grown in the dark for seven days and subsequently transferred to FR light (5 μmol·m$^{-2}$·s$^{-1}$), R light (17.56 μmol·m$^{-2}$·s$^{-1}$), B light (13 μmol·m$^{-2}$·s$^{-1}$), or W light (85 μmol·m$^{-2}$·s$^{-1}$) for 4 h. The values represent mean ± SEM of three replicates. The significant differences between data were calculated using Student's *t*-test, and indicated with asterisks (*$P < 0.05$, ** $P < 0.01$, *** $P < 0.001$).

*Brassica napus* [41] is Dehydrin and LEA4, respectively. The distributions of *LEA* family members in different species indicate that *LEA* genes play unique roles in stress response and regulation of plant growth and development in different species. These differences also suggest that the *LEA* family genes may have evolved independently after the divergence of these species.

Normally, the diversity of structures and conserved domains caused the evolution of multigene families [42]. Our analysis revealed that the majority of LEA proteins in the same group shared similar domains, suggesting that these conserved domains may play crucial roles in group-specific functions. However, high divergence was also found in the structures between different groups. Identical results were reported earlier for LEA proteins in *Arabidopsis* [43], rice [5], and tomato (*Solanum lycopersicon* L.) [44]. Rye has the largest genome (7.86 Gbp) among all diploid species in Triticeae, with more than 90% repetitive sequences [45]. Gene duplication is considered to be one of the primary driving forces in the evolution of genomes

and genetic systems [46]. Twenty-eight *ScLEA* genes were clustered into 12 tandem duplication blocks, and 27 *ScLEA* genes were clustered into 19 segmental duplication blocks in rye. These results indicate that members of the same group may have originated via gene duplication, and that different groups may have evolved from different ancestral genes with various domain structures [47].

## Expression analysis and functional prediction of rye *LEA* genes

Analysis of expression profiles in different tissues revealed obvious temporal and spatial specificity in *ScLEA* genes expression, with the highest expression for most genes mainly observed in grains (Fig 5). The expression levels of *LEA* genes in *Arabidopsis* [4] and wheat [48] were also reported to be generally higher in seeds. Transcriptome data for rye subjected to drought stress showed that, except for *ScDehydrin-3*, all Dehydrin group genes were responsive to drought treatment (S1 Fig). Dehydrin group genes were also specifically and highly expressed in grains (Fig 5). These results showed that the expression levels of Dehydrin group genes are upregulated during seed maturation, suggesting that the expression of these genes might be induced by dehydration at the later stage of seed maturation [48]. The correlation between LEA protein and desiccation tolerance had been confirmed in many orthodox seeds [24,49]. During the mature dehydration of seeds, the Dehydrin protein will be present as space filler in cells, maintaining the dissolved character of cell fluid to avoid damage to cell architecture [15]. The molecular function of Dehydrin proteins in the process of seed maturation will provide new ideas for the genetic improvement of crops.

Light affects plant growth and developmental processes including seed germination, dormancy, circadian rhythms, flower induction, plant architecture, and shade avoidance [50]. Studies have found that LEA protein expression in *Escherichia coli* promotes tolerance to UV radiation [51,52]. At present, studies of LEA proteins in plants generally focus on drought, salt, plant hormone, and low temperature stresses, including UV radiation stress, but few studies have been conducted on the response to different light qualities. Our results showed that *ScLEA* genes responded to FR, R, B, and W light, and the transcription level was almost the highest under blue light. Thus, *ScLEA* genes may play certain roles in blue light signal, and those finding may provide a new perspective for the future study of new functions of ScLEA proteins. Rye is mainly grown in northern Europe and North Africa [53], and the mountainous or colder northern parts of China. Most of the regions experience a wide range of temperatures, altitudes, rainfall, and light levels. Therefore, the analysis of different stress treatments in this study lays a foundation for exploring the function of LEA proteins in improving the stress tolerance of rye.

## *Cis*-acting elements involved in the mediation of rye *LEA* gene expression

Plants frequently adapt to environmental stresses by regulating the rate of transcription [54]. *Cis*-acting elements of genes are binding sites for transcription factors that activate or repress transcription [55]. Some promoter elements, such as the TATA, GC, and CCAAT boxes, are common to many genes. In response to abiotic stresses such as heat shock and hormones, a large number of specific elements are also involved in transcriptional regulation [55]. In *Arabidopsis*, most genes encoding LEA proteins were found to have ABRE and LTRE elements in their promoters and many genes containing these elements were induced by ABA, cold, or drought treatment [4]. In this study, we also found ABRE and LTRE elements in the *ScLEA* promoters (S4 Fig). The results of qRT-PCR showed that the expression of 12 *ScLEA* genes was induced by ABA treatment and low temperature, and that the expression levels were significantly upregulated with a decrease of temperature. Moreover, light responsive elements were

also found in the *ScLEA* promoter, and *ScLEA* genes responded to different light qualities. Therefore, *cis*-acting elements that may influence the expression of *ScLEA* should also be investigated when analyzing the role of *ScLEA* genes under various stresses.

## Conclusion

LEAs are important proteins that respond to biotic and abiotic stresses and have been described in numerous plants. In this study, a total of 112 *ScLEA* genes were identified in rye and classified into eight groups. All *ScLEA* genes contained few or no introns, and all encoded proteins containing the conserved LEA domain. *ScLEA* genes were distributed on all rye chromosomes with some clustering. By analyzing promoter elements in combination with RNA-seq data and qRT-PCR results, we showed that *LEA* genes in rye are strongly responsive to abiotic stresses such as drought, low temperature, light quality, ABA, and NaCl. These results lay a foundation for further investigating the functions of LEA proteins and their potential use in genetic improvement of crops.

## Supporting information

**S1 Fig. RNA-seq expression profiles of 112 *ScLEA* genes after drought treatment for 0 h, 3 h, 6 h, and 12 h.** The heatmap was constructed using TBtools. The color scale on the right represents relative expression levels: red represents high level and blue represents low level. (TIF)

**S2 Fig. The relative expression levels of six *ScLEA* genes under different abiotic stresses were analyzed by qRT-PCR.** Ten-day-old seedling leaves were sampled after 0 h, 3 h, 6 h, 9 h, 12 h, and 24 h under 20% PEG6000, 100 μM ABA, 200 mM NaCl, or 100 mM mannitol. The values represent mean ± SEM of three replicates. The significant differences between data were calculated using Student's *t*-test, and indicated with asterisks (*$P$ <0.05, ** $P$<0.01, *** $P$<0.001).
(TIF)

**S3 Fig. RNA-seq expression profiles of 112 *ScLEA* genes after incubation at -10˚C for 0 h, 1 h, 4 h, and 8 h.** The heatmap was constructed using TBtools. The color scale on the right represents relative expression levels: red represents high level and blue represents low level.
(TIF)

**S4 Fig. Predicted *cis*-acting elements in the promoters of 12 *ScLEA* genes.**
(TIF)

**S1 Table. List of qRT-PCR primers for the 12 *ScLEA* genes in this study.**
(XLSX)

**S2 Table. List of the 112 *ScLEA* genes identified in this study.**
(XLSX)

**S3 Table. Tandem duplicated *ScLEA* genes.**
(XLSX)

**S4 Table. Segmental duplicated *ScLEA* genes.**
(XLSX)

## Author Contributions

**Conceptualization:** Jianping Yang.

**Formal analysis:** Mengyue Ding, Weimin Zhan.

**Investigation:** Guanghua Sun.

**Methodology:** Xiaolin Jia, Shizhan Chen, Wusi Ding.

**Writing – original draft:** Mengyue Ding, Lijian Wang.

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
