## [Decision Letter · Decision Letter 0]

19 Feb 2021

PONE-D-21-02217

Genome-wide identification and expression analysis of late embryogenesis abundant protein-encoding genes in rye (Secale cereale L.)

PLOS ONE

Dear Dr. Wang,

Thank you for submitting your manuscript to PLOS ONE. After careful consideration, we feel that it has merit but does not fully meet PLOS ONE’s publication criteria as it currently stands. Therefore, we invite you to submit a revised version of the manuscript that addresses the points raised during the review process.

We look forward to receiving your revised manuscript.

Kind regards,

Aimin Zhang, Ph.D.

Academic Editor

PLOS ONE

Journal Requirements:

2. We note that you are reporting an analysis of a microarray, next-generation sequencing, or deep sequencing data set. PLOS requires that authors comply with field-specific standards for preparation, recording, and deposition of data in repositories appropriate to their field. Please upload these data to a stable, public repository (such as ArrayExpress, Gene Expression Omnibus (GEO), DNA Data Bank of Japan (DDBJ), NCBI GenBank, NCBI Sequence Read Archive, or EMBL Nucleotide Sequence Database (ENA)). In your revised cover letter, please provide the relevant accession numbers that may be used to access these data. For a full list of recommended repositories, see http://journals.plos.org/plosone/s/data-availability#loc-omics or http://journals.plos.org/plosone/s/data-availability#loc-sequencing

Reviewers' comments:

Reviewer's Responses to Questions

**Comments to the Author**

1. Is the manuscript technically sound, and do the data support the conclusions?

Reviewer #1: Yes

2. Has the statistical analysis been performed appropriately and rigorously? 

Reviewer #1: No

3. Have the authors made all data underlying the findings in their manuscript fully available?

Reviewer #1: Yes

4. Is the manuscript presented in an intelligible fashion and written in standard English?

Reviewer #1: Yes

5. Review Comments to the Author

Reviewer #1: This study identified 112 late embryogenesis abundant (LEA) genes in Secale cereale. These genes were subjected to structure analysis. Temporal and spatial specificity of some ScLEA genes were studied. Expression of several genes were induced by different PEG treatment, low temperature, and blue light. Results of this study may be useful for the ScLEA genes in stress tolerance. The study was well designed and the manuscript was well prepared.

Specific comments:

qPCR analysis: sampling strategy and statistical analysis must be described in the M&Ms.

Results: avoid literature citation in the Results section.

Other minor suggestions are marked in the edited manuscript.

6. PLOS authors have the option to publish the peer review history of their article (what does this mean?). If published, this will include your full peer review and any attached files.

Reviewer #1: No

---

## [Author Response · Author response to Decision Letter 0]

9 Mar 2021

Dear Reviewer 1，

Thanks for your comments on our paper. We have revised our paper according to your comments:

1. qPCR analysis: sampling strategy and statistical analysis must be described in the M&Ms.

Answer: Sampling strategy and statistical analysis have been added in the M&Ms.

2. Results: avoid literature citation in the Results section.

Answer: References cited in the results section, transferred to the introduction or methods section, or deleted.

3. Other minor suggestions are marked in the edited manuscript.

Answer: Adoption of reviewer suggestions.

---

## [Decision Letter · Decision Letter 1]

25 Mar 2021

Genome-wide identification and expression analysis of late embryogenesis abundant protein-encoding genes in rye (Secale cereale L.)

PONE-D-21-02217R1

Dear Dr. Wang,

We’re pleased to inform you that your manuscript has been judged scientifically suitable for publication and will be formally accepted for publication once it meets all outstanding technical requirements.

Kind regards,

Aimin Zhang, Ph.D.

Academic Editor

PLOS ONE

Additional Editor Comments (optional):

Reviewers' comments:

Reviewer's Responses to Questions

**Comments to the Author**

1. If the authors have adequately addressed your comments raised in a previous round of review and you feel that this manuscript is now acceptable for publication, you may indicate that here to bypass the “Comments to the Author” section, enter your conflict of interest statement in the “Confidential to Editor” section, and submit your "Accept" recommendation.

Reviewer #1: All comments have been addressed

2. Is the manuscript technically sound, and do the data support the conclusions?

Reviewer #1: Yes

3. Has the statistical analysis been performed appropriately and rigorously? 

Reviewer #1: Yes

4. Have the authors made all data underlying the findings in their manuscript fully available?

Reviewer #1: Yes

5. Is the manuscript presented in an intelligible fashion and written in standard English?

Reviewer #1: Yes

6. Review Comments to the Author

Reviewer #1: A few more are listed below: 1. Please change add 'L.' after Gossypium hirsutum. 2. Delete phase before duration. 3. Change Student's t-test or two way ANOVA to the Student's t-test or the two-way analysis of variance (ANOVA).

7. PLOS authors have the option to publish the peer review history of their article (what does this mean?). If published, this will include your full peer review and any attached files.

Reviewer #1: **Yes: **Hongjie Li

---

## [Editor Report · Acceptance letter]

1 Apr 2021

PONE-D-21-02217R1 

Genome-wide identification and expression analysis of late embryogenesis abundant protein-encoding genes in rye (*Secale cereale* L.) 

Dear Dr. Wang:

I'm pleased to inform you that your manuscript has been deemed suitable for publication in PLOS ONE. Congratulations! Your manuscript is now with our production department. 

Kind regards, 

on behalf of

Prof. Aimin Zhang 

Academic Editor

PLOS ONE